# Novel Genetic Microvascular Dysplasia Causing Hypoperfusion of Cardiac, Renal, and Cerebral Circulation

**DOI:** 10.3390/jcm12227150

**Published:** 2023-11-17

**Authors:** Andrea Frustaci, Rosario Cianci, Romina Verardo, Bruna Cerbelli, Maria Cecilia D’Asdia, Alessandro De Luca

**Affiliations:** 1IRCCS San Raffaele, Via di Val Cannuta, 247, 00166 Rome, Italy; 2Head of Nephrological “Dh and Day-Service Unit”, Policlinic Umberto I, Sapienza University of Rome, 00161 Rome, Italy; rosario.cianci@uniroma1.it; 3Cellular and Molecular Cardiology Lab, IRCCS L. Spallanzani, 00149 Rome, Italy; romina.verardo@inmi.it; 4Department of Medico-Surgical Sciences and Biotechnology, Sapienza University of Rome, 00161 Rome, Italy; bruna.cerbelli@uniroma1.it; 5Medical Genetics Division, Fondazione IRCCS Casa Sollievo della Sofferenza, 71013 San Giovanni Rotondo, Italy; m.dasdia@css-mendel.it (M.C.D.); a.deluca@css-mendel.it (A.D.L.)

**Keywords:** microvascular dysplasia, genetic, angina, ischemia

## Abstract

Background: Microvascular disorders represent an uncommon site of tissue hypo-perfusion and damage. Various genetic and acquired causes can be involved. A 65-year-old man was admitted because of refractory angina, which he had had since the age of 30 years, micro-hematuria, and recurrent transitory ischemic attacks from the age of 64. Methods: Hematochemical studies, ECG, Holter monitoring, 2D-echo, cardiac magnetic resonance (CMR), CTA of cerebral vessels, endomyocardial coronary angiography, and kidney biopsy processes were undertaken. Gene mutation analysis was conducted using next-generation sequencing, which included more than 5000 genes associated with inherited diseases. Results: Hematochemical findings were unremarkable. The ECG, Holter, 2D-echo, and CTA of brain vessels were normal. Cerebral magnetic resonance showed the presence of multiple small foci of ischemia. Coronary and ventricular angiography showed normal arteries with remarkably slow flow and multiple biventricular micro-aneurysms. At the endomyocardial biopsy, five of seven arterioles presented severe lumen obstruction due to hypertrophy and disarray of the muscular coat. Similarly, obstructed pre-glomerular arteries with glomerular sclerosis were seen at the renal biopsy. Genetics identified mutations in the ABCC6, MMP2, and XYLT1 genes, which play pivotal roles in the extracellular matrix. Conclusion: This study described a new genetic microvascular obstructive disease causing progressive hypo-perfusion of the human brain, heart, and kidney.

## 1. Introduction

Angina reflects chest pain due to myocardial ischemia where the structural context is extremely variable and may affect the patient’s treatment and outcome. Indeed, angina may derive from anomalies of the origin and course of coronary arteries often susceptible to surgical repair [1], atherosclerotic lesions of epicardial vessels usually resulting from PTCA and stenting and/or bypass [2,3], myocardial bridges and tunnels [4,5], potentially suitable for surgical correction [5], and small vessel disease [6], where the therapeutic impact is frequently disappointing. In the last instance, intramural vessels can be obstructed because of heritable diseases including sarcomeric diseases, such as hypertrophic cardiomyopathy, or non-sarcomeric gene mutations, as in Fabry disease [7], or affected by acquired microvascular dysfunction, as in coronary syndrome x [8]. However, despite major diagnostic efforts, angina may have a difficult pathophysiological definition. The following report described the case of persistent angina, refractory to high-dose calcium antagonists, nitrates, and beta-blockers due to microvascular coronary dysplasia. Cardiac affection is associated with similar microvascular obstruction causing renal microhematuria and recurrent transitory cerebral ischemic attacks. An extensive genetic study identified genetic variants in the ABCC6, MMP2, and XYLT1 genes.

### Case Study

A 65-year-old man had, in the last year, several hospital admissions because of transitory ischemic brain attacks characterized with CTA and magnetic resonance by multiple small foci of cerebral ischemia with normal carotid, vertebral, and main cerebral arteries, requiring the administration of anti-platelet aggregation drugs, nimodipine, nitrates, anticoagulants, and mannitol or glycerol infusion to contrast cerebral edema, headache, and neurologic symptoms. The patient was normotensive and devoid of metabolic diseases, including diabetes and dyslipidemias, and had no history of smoking. ECG was normal and Holter monitoring failed to show arrhythmias. The echocardiogram and CMR showed normal cardiac dimension and biventricular contractility while failing to show the presence of cardiac thrombi. His history started at 30 years of age with spontaneous and effort angina, which was investigated with multiple coronary angiographies showing systematically normal coronary arteries with remarkably slow flow (Appendix A) in the absence of systemic artery hypertension and valvular heart disease. The echocardiogram showed normal cardiac parameters with preserved right and left ventricular function. Cardiac magnetic resonance showed normal LV wall thickness (the maximal wall thickness was 10 mm) and only a small area of positive LGE, compatible with fibrous replacement, was found in the left inferior wall. An electrocardiographic stress test induced chest pain at a heart rate of 120 bts/min without any significant change in ST segment and T wave, while stress ^99m^Tc myocardial perfusional scintigraphy was negative. Following coronary sinus catheterization and atrial pacing, the occurrence of chest pain at 120 bts/min was associated with a clear inversion of lactates from the aorta and coronary sinus blood samples (from 1/0.5 to 1/1.2 mg/dL), suggesting chest pain to be due to myocardial ischemia. When he was forty, following a prolonged angina episode, ventriculography revealed the occurrence of micro-aneurysms at the base of the right ventricle and in the left ventricular free wall. At that time, an invasive study was implemented with bi-ventricular endomyocardial biopsy. At histology, no evidence of myocarditis, infiltrative, or storage disease was documented, and PCR for the most common cardiotropic viruses was negative. Major abnormalities were documented in five of seven arterioles included in six biopsy fragments, showing a severe lumen obstruction due to an abnormal thickening of the muscular coat. The arterioles’ middle layer presented hypertrophy with hyperplasia and disorganization of smooth muscle cells that were focally replaced by fibrous tissue, denoting a dysplastic process (Figure 1). A prominence of adventitial ganglia (Figure 1) could have had a role in the hypertrophy and disarray of smooth muscle cells of the media through the release of trophic mediators (Cathecolamines?).

The anti-myocardial ischemia drug regimen included nitrates, beta-blockers, Ca antagonists, and aspirin, obtaining transitory attenuation of the anginal symptoms. At the age of 60 years, the patient complained of recurrent hematuria with normal values of renal echo and creatinine clearance. A renal biopsy manifested histological changes similar to endomyocardial findings, consisting of medial thickening of small and medial-sized renal arteries associated with focal glomerular sclerosis, and interstitial and replacement fibrosis (Figure 2).

In order to identify possible gene variants associated with this condition, we used the SureSelect Custom Constitutional Panel 17Mb (CCP17) (Agilent, Santa Clara, CA, USA), a targeted next-generation sequencing (NGS) panel for inherited disease comprising more than 5000 genes, and selected those variants located in genes related to the patient’s clinical features, as they are defined by the “Human Phenotype Ontology, HPO”, which provides a standardized vocabulary of phenotypic abnormalities encountered in human disease. The following HPO terms were used: “hereditary hemorrhagic telangiectasia”, “congenital vascular malformation”, “vascular smooth muscle hypertrophy”, “microhematuria”, “small vessel systems disease”, “transient ischemic attack”, “abnormal cerebral vascular morphology”, “abnormal renal vascular morphology”, “arteriovenous malformation”, and “vascular systems disease”. Of note, bioinformatic analysis identified three functionally related variants, as they affected genes that play pivotal roles in the extracellular matrix. The first variant was a heterozygous missense substitution (NM_001171.6:c.1171A>G; p.Arg391Gly) in the ABCC6 gene. This variant was classified as a variant of uncertain significance (VUS) because it met two pathogenic supporting criteria of the American College of Medical Genetics and Genomics-Association for Molecular Pathology (ACMG-AMP), namely computational predictions (PP3) and localization in a mutational hot-spot where benign missense variants are not described (PM1). The ABCC6 gene encodes an ATP-binding cassette (ABC) protein, which functions as an efflux transporter. Its specific substrate remains unknown, although it has been recently demonstrated that ABCC6 plays a significant role in the regulation of plasmatic inorganic pyrophosphate, a potent inhibitor of ectopic mineralization. The second variant was a missense substitution in the MMP2 gene (NM_004530.6:c.1483C>T; p.Arg495Trp), which encodes a member of the matrix metalloproteinase gene family, which are zinc-dependent enzymes capable of cleaving components of the extracellular matrix. According to the ACMG-AMP criteria, this variant was classified as VUS because the PM2 pathogenic moderate criterion (extremely low frequency in the gnomAD population database: 0.003%) (gnomAD: https://gnomad.broadinstitute.org/ accessed on 9 May 2023) was met. The third variant was a missense change in the XYLT1 gene (NM_022166.4:c.2081G>A; p.Arg694His), which encodes an enzyme that catalyzes the transfer of UDP-xylose to serine residues of a substrate acceptor protein, a transfer reaction necessary for the biosynthesis of glycosaminoglycan chains. According to the ACMG-AMP criteria, this variant was also classified as VUS because it met the pathogenic moderate criterion PM2, and the benign supporting criterion BP4, referred to as bioinformatic predictions. Family history was negative and unaffected family members were unavailable for genetic testing.

## 2. Materials and Methods

### 2.1. Cardiac Studies

Cardiac investigations, including noninvasive (ECG, Holter monitoring, 2D-echocardiography, and CMR) and invasive (coronary, left ventricular angiography, coronary sinus catheterization with atrial pacing, and left ventricular EMB) procedures, were performed after receiving written informed consent. EMB is regularly performed in our institution whenever a symptomatic heart muscle disease remains undiagnosed by noninvasive procedures including echocardiography and cardiac magnetic resonance. Biopsy samples, 5–8 fragments, were cut and stored at −80 °C, or processed for histology, immunohistochemistry, and electron microscopy. Two frozen samples were processed for real-time PCR for the most common cardiotropic viruses in case of an observation of overlapping myocarditis at histology.

This study complies with the Declaration of Helsinki, the locally appointed ethics committee (opinion number 6/2019) approved the research protocol, and informed consent was obtained from the patient. The patient was strongly motivated to clarify the origin of symptoms and gave their written consent for the procedure.

Pathologists, echocardiographers, and CMR investigators were blind to the patient’s clinical and genetic background.

### 2.2. Cardiac Magnetic Resonance

CMR exams were performed on a 1.5 Tesla scanner (Avanto, Siemens, Rome, Italy). The CMR protocol included (i) cine CMR sequence acquired during breath holds in the short-axis, 2-chamber, and 4-chamber; (ii) black blood T2-weighted short tau inversion recovery images on short-axis planes covering the entire left ventricle during 6 to 8 consecutive breath holds for myocardial edema detection; (iii) late gadolinium-enhanced imaging performed 15 min after injection of 0.2 nmol/kg of gadoterate meglumine, and signal intensity value 2 SDs above the mean signal intensity of the remote normal myocardium was considered suggestive of myocardial fibrosis; (iv) native T1 mapping imaging was performed, when available, using the MOLLI sequence on three short-axis views (one basal and two midventricular); (v) a T2 map was obtained using a T2-prepared True-FISP prototype sequence producing 3 single-shot images with 3 different T2 pulse preparations. A nonrigid registration algorithm and two-parametric automatic curve fitting were automatically applied to generate the map. CMR image analysis was performed as previously described, extending the analysis method for T1 maps to T2 maps. In particular, the values of T1 and T2 global were defined as normal, reduced, or increased compared to a reference range developed on a multiage sample of 100 healthy subjects of both sexes (normal value nT1 < 970 ms, T2 > 49.7 ms).

### 2.3. Invasive and Endomyocardial Biopsy Studies

Cardiac catheterization with left ventricular and coronary angiography was obtained. EMB was performed in the septal–apical region of the left ventricle. Endomyocardial samples were blindly evaluated by the same pathologist, who was informed of the clinical and genetic characteristics after morphological examination.

### 2.4. Histology and Immunohistochemistry

For histological analysis, the endomyocardial samples were fixed in 10% buffered formalin and paraffin-embedded. Five-micron-thick sections were stained with hematoxylin and eosin and Masson trichrome.

The histological diagnosis of myocarditis was based on the evidence of leukocyte infiltrates (≥14 leukocytes/2 mm) associated with necrosis of the adjacent myocytes, according to the Dallas criteria confirmed by immunohistochemistry. In particular, for the phenotypic characterization of inflammatory infiltrates, immunohistochemistry was performed for CD3, CD20, CD43, CD45RO, and CD68 (all Dako, Carpinteria, CA, USA).

### 2.5. Molecular Study

The definition of virus-negative, immune-mediated, overlapping myocarditis followed the presence at histology of ≥7 CD3+ T lymphocytes per low-power field associated with focal necrosis of the adjacent myocytes. The molecular study was performed in the patient with Real-Time PCR for the most common cardiotropic viruses (Adenovirus, Enterovirus, Influenza A and B viruses, Epstein Barr virus, Parvovirus B19, Hepatitis C virus, Cytomegalovirus, Human Herpes virus 6, and Herpes Simplex types 1 and 2) for the possible identification of viral genomes.

CTA and magnetic resonance were obtained to investigate the cerebral transitory ischemic attacks. A renal biopsy was also undertaken because of the presence of microhematuria.

## 3. Genetic Study

### 3.1. DNA Extraction

Genomic DNA was extracted from peripheral blood using a manual kit (Macherey-Nagel, Duren, Germany) according to the manufacturer’s instructions. DNA concentration was assessed throughout using a Qubit™ fluorometer (Invitrogen, Carlsbad, CA, USA), and purity parameters (260/280 = 1.8/2; 230/280 = 1.8/2) were evaluated using the NanoDrop1000 Spectrophotometer (Thermo Scientific, Waltham, MA, USA).

### 3.2. SureSelect Custom Constitutional Panel 17Mb (CCP17) Target Enrichment Library Preparation

Targeted enriched libraries were prepared using the SureSelect Custom Constitutional Panel 17 Mb (CCP17), a targeted NGS panel designed to capture all the coding exons and intronic flanking regions (±25 bp) of more than 5000 genes associated with inherited diseases (Agilent Technologies, Santa Clara, CA, USA).

In detail, a genomic DNA sample was suspended in nuclease-free water to a final concentration of 25 ng/µL and final volume of 2 µL and enzymatically fragmented to achieve a target peak positioned between 245 and 325 bp. Agilent’s SureSelect Custom Constitutional Panel 17Mb (CCP17) Target Enrichment was used for library preparation according to the manufacturer’s recommendations. Ten cycles of PCR were performed for the amplification of the post-captured library, and the quality of the final DNA library was assessed using the High Sensitivity Agilent 2100 Bioanalyzer System (Agilent Technologies, Santa Clara, CA, USA). Per the manufacturer’s protocol, the library peak size was in the range of 325 and 450 bp. Samples were then pooled and sequenced using the NextSeq Mid Output Kit v2.5 (300 Cycles) (Illumina, San Diego, CA, USA) on NextSeq 500 (Illumina, San Diego, CA, USA) according to the manufacturer’s recommendations for paired-end 150-bp reads.

### 3.3. Variant Annotation, Filtering, Prioritization, and Classification

Generated reads were aligned to the human genome reference sequence (assembly GRCh37/hg19) using Bowtie 2 (version 2.3.0). BAM files were sorted using the SAM tools (version 1.3.2) and purged from candidate PCR duplicates using the Mark Duplicates tool from the Picard suite (version 2.9.0). The local realignment and base-quality-score recalibration functions were performed using the Genome Analysis Toolkit (GATK 4.0). Reads with mapping quality scores lower than 20 were filtered out. The GATK’s Haplotype Caller was used to identify single-nucleotide polymorphisms and insertions/deletions. Genetic variants were annotated using ANNOVAR and filtered. Common variants with a minor allele frequency (MAF) >1% in dbSNP [https://www.ncbi.nlm.nih.gov/projects/SNP/ accessed on 9 May 2023], GO-ESP [https://esp.gs.washington.edu/drupal/ accessed on 9 May 2023] and GnomAD [http://gnomad.broadinstitute.org/ accessed on 9 May 2023] databases were excluded. Variants’ phenotype-based prioritization of candidate genes was conducted using Geneyx Analysys Software Version 5.10 (Geneyx Genomex, Herzliya, Israel). Validation and segregation analyses were performed by Sanger sequencing using the ABI Prism BigDye Terminator v3.1 Cycle Sequencing Kit (Applied Biosystems, Foster City, CA, USA). Sanger sequencing results were analyzed using Mutation Surveyor V5.0.0 software (SoftGenetics, State College, PA, USA). Variants’ classifications were determined using the American College of Medical Genetics and Genomics-Association for Molecular Pathology (ACMG-AMP) standards and guidelines for the interpretation of sequence variants [9]. ACMG-AMP pathogenicity criteria were assigned using Franklin software, a comprehensive online interpretation tool from Genoox (https://franklin.genoox.com/clinical-db/home accessed on 9 May 2023), and manually modified if not applicable to the variant under investigation.

## 4. Results

A 2-D echocardiogram failed to show abnormal findings, including the presence of mitral prolapse and pericardial effusion; left ventricular dimensions and contractility (EF 60%) were normal. An electrocardiographic stress test induced chest pain at a heart rate of 120 bts/min without any significant change in the ST segment and T wave, while stress 99 mTc myocardial perfusional scintigraphy was negative. Following coronary sinus catheterization and atrial pacing, the occurrence of chest pain at 120 bts/min was associated with a clear inversion of lactates from the aorta and coronary sinus blood samples (from 1/0.5 to 1/1.2 mg/dL), suggesting chest pain to be due to myocardial ischemia. At cardiac magnetic resonance, LV wall thickness was normal and only a small area of fibrous replacement was appreciated in the left inferior wall. A new coronary angiograph showed no obstructive lesions but a very slow coronary flow (see Appendix A). In addition, at RV and LV angiography, several microaneurysms were visualized in the inferior segment of LV and the posterior portion of RV (Figure 1). For the histology of endomyocardial samples, no evidence of myocarditis, infiltrative, or storage disease was documented. Cardiomyocytes were mildly hypertrophied, measuring at nuclear level 18 ± 0.8 microns and regularly arranged, ruling out a possible diagnosis of hypertrophic cardiomyopathy. The PCR for the most common cardiotropic viruses (including adeno and enteroviruses, EBV, HCV, Influenza A/B, cytomegalovirus, and Parvovirus B19) was negative. Major abnormalities were documented in five of seven arterioles included in six biopsy fragments. In particular, the five arterioles involved, of 48, 52, 58, 62, and 66 microns in diameter, presented a severe lumen obstruction due to an abnormal thickening of the muscular coat. The arterioles’ middle layer presented hypertrophy with hyperplasia and a disorganization of smooth muscle cells that were focally replaced by fibrous tissue, also denoting a dysplastic process (Figure 1). One affected arteriole showed a prominent ganglion (G) in the adventitia (Figure 1).

CTA and brain magnetic resonance showed multiple small foci of cerebral ischemia with normal carotid, vertebral, and main cerebral arteries. A histology of the renal biopsy showed an obstruction of the renal arterioles associated with sclerosed glomeruli and renal fibrosis (see Figure 2).

## 5. Genetic Results

Bioinformatic analysis identified three functionally related variants, as they affected genes that play pivotal roles in the extracellular matrix. The first variant was a heterozygous missense substitution (NM_001171.6:c.1171A>G; p.Arg391Gly) in the ABCC6 gene. The ABCC6 gene encodes an ATP-binding cassette (ABC) protein, which functions as an efflux transporter. Its specific substrate remains unknown, although it has been recently demonstrated that ABCC6 plays a significant role in the regulation of plasmatic inorganic pyrophosphate, a potent inhibitor of ectopic mineralization. The second variant was a missense substitution in the MMP2 gene (NM_004530.6:c.1483C>T; p.Arg495Trp), which encodes a member of the matrix metalloproteinase gene family, which are zinc-dependent enzymes capable of cleaving components of the extracellular matrix. The third variant was a missense change in the XYLT1 gene (NM_022166.4:c.2081G>A; p.Arg694His), which encodes an enzyme that catalyzes the transfer of UDP-xylose to serine residues of a substrate acceptor protein, a transfer reaction necessary for the biosynthesis of glycosaminoglycan chains.

## 6. Discussion

The present case study reported a novel genetic entity consisting of microvascular obstructive disease causing progressive hypo-perfusion of the human brain, heart, and kidney. The demonstration is based on manifestations of angina early in life (30 years), proved by the detection of increased lactate production from the coronary sinus during atrial pacing in the absence of an obstruction of the epicardial coronary arteries and remarkably slow flow at coronary angiography. With advancing age, angina episodes became more frequent and stronger, and in a biventricular control, angiography was associated with the occurrence of biventricular micro-aneurysms (see Figure 1). Of interest, ventricular microaneurysms can be missed by cardiac magnetic resonance while being easily detectable with ventricular angiography, as previously reported [10]. Their occurrence usually complicates an inflammatory myocardial disease and is often associated with preserved cardiac dimensions and function [10]. Their major clinical manifestations are represented by sometimes life-threatening ventricular arrhythmias, so their recognition is necessary for appropriate treatment.

To clarify the structural abnormalities of intramural vessels, as well as their pathogenetic mechanisms, a left ventricular endomyocardial biopsy has been undertaken following the patient’s agreement and consent of the local ethical committee. This investigation was necessary as many congenital [11], lysosomal storage disorders, such as Anderson–Fabry disease, sarcomeric hypertrophic cardiomyopathy [7], and even acquired causes such as anthracycline damage [12], as well as infectious effects on small coronary vessels, can be involved. In particular, Syndrome X can be sustained by endotheliotropic viral agents [8] including adenovirus, EBV, HHV6, and mostly Parvovirus B19. In Fabry cardiomyopathy a small coronary vessel disease is promoted by hypertrophy and hyperplasia of vessels’ smooth muscle cells that appear to contain vacuoles with large amounts of glycosphingolipids due to deficiency of the lysosomal enzyme alpha-galactosidase A. Microvascular involvement may occur very early, affecting the pre-hypertrophic phase of the disease, and can be limited or electively involve the coronary intramural vessels. In our case, affected vessels presented non-vacuolated smooth muscle cells, excluding that instance, and, in general, storage diseases. Indeed, in our case, most coronary arterioles (≤100 microns in diameter) were remarkably obstructed because of the thickening of the vessel media determined by hypertrophy with hyperplasia and the disarray of smooth muscle cells associated with replacement fibrosis. In the advent of an affected arteriole, a hypertrophied ganglion was documented, raising the hypothesis of their possible pathogenic implications through the release of hormonal mediators like catecholamines. These aspects depict a dysplastic process and raise the possibility of a genetic disorder. The clinical impact was represented by the generation of various biventricular microaneurysms visualized in the right and left ventriculography that represent a major risk of cardiac wall rupture, as well as of electrical instability. Treatment with anti-ischemic drugs, such as beta-blockers, nitrates, aspirin, and calcium antagonists, attenuated the symptomatology, which had several peaks of recurrence.

A second level of clinical manifestation was the kidney with recurrent micro-hematuria, which was unexplained at non-invasive investigations, including renal echography and a doppler analysis of renal arteries, as well as renal function tests; therefore, a renal biopsy was required. Renal histology showed similar morphologic changes to those revealed at the endomyocardial biopsy. They consisted of obstructions of pre-glomerular arterioles because of medial hypertrophy associated with glomerular sclerosis and diffuse areas of fibrous replacement. No inflammatory infiltrates or cell vessel inclusions were documented, similar to the endomyocardial biopsy findings.

Finally, at the age of 64 years, our patient manifested recurrent headaches with disorientation and syncope. These symptoms were associated with several ischemic cerebral foci at magnetic resonance in the absence of abnormalities of carotid, vertebral, and major brain arteries. These transitory ischemic attacks required frequent mannitol and glycerol infusion to recover from cerebral edema and neurologic manifestations.

Genetic analysis identified three functionally related variants as the affected genes that play pivotal roles in the extracellular matrix. The first variant was a heterozygous missense substitution (NM_001171.6:c.1171A>G; p.Arg391Gly) in the ABCC6 gene. The ABCC6 gene encodes an ATP-binding cassette (ABC) protein, which functions as an efflux transporter. Its specific substrate remains unknown, although it has been recently demonstrated that ABCC6 plays a significant role in the regulation of plasmatic inorganic pyrophosphate, a potent inhibitor of ectopic mineralization. The second variant was a missense substitution in the MMP2 gene (NM_004530.6:c.1483C>T; p.Arg495Trp), which encodes a member of the matrix metalloproteinase gene family, which are zinc-dependent enzymes capable of cleaving components of the extracellular matrix. The third variant was a missense change in the XYLT1 gene (NM_022166.4:c.2081G>A; p.Arg694His), which encodes an enzyme that catalyzes the transfer of UDP-xylose to serine residues of a substrate acceptor protein, a transfer reaction necessary for the biosynthesis of glycosaminoglycan chains.

Although a cause–effect relationship is not provided, there are several pieces of pathogenic evidence correlating these three variants. Recessive loss-of-function variants of the ABCC6 gene [13,14] cause pseudoxanthoma elasticum, a multisystemic condition characterized by the calcification and fragmentation of elastic fibers in the skin, eyes, and the cardiovascular system. Elevated production of MMP2 in pseudoxanthoma elasticum fibroblasts and increased levels of MMP2 were observed in serum from pseudoxanthoma elasticum patients. Moreover, variations in the MMP2 gene were found as a genetic co-factor for pseudoxanthoma elasticum [15]. Furthermore, high XYLT1 activity was found in the sera of affected patients with pseudoxanthoma elasticum, reflecting a higher rate of proteoglycan biosynthesis in these patients.

## 7. Conclusions

This report described a new genetic microvascular obstructive disease causing progressive hypo-perfusion of the human brain, heart, and kidney. Mutation analysis conducted using next-generation sequencing through the SureSelect Custom Constitutional Panel 17Mb (CCP17), which includes more than 5000 genes associated with inherited diseases, identified genetic variations in the ABCC6, MMP2, and XYLT1 genes.

## Figures and Tables

**Figure 1 jcm-12-07150-f001:**
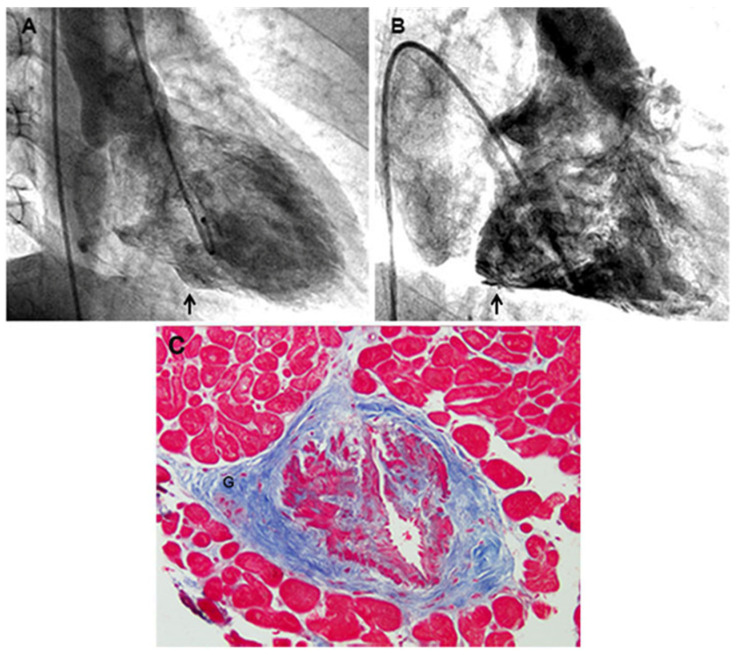
(**A**) Presence of a micro-aneurysm in the inferior LV wall (arrow). (**B**) Evidence of micro-aneurysms in the posterior RV free wall (arrow). (**C**) Hypertrophy, hyperplasia, and disorganization of smooth muscle cells, focally replaced by fibrous tissue, are observed (Masson Trichrome, 200×). A prominent ganglion is shown in the adventitia (G).

**Figure 2 jcm-12-07150-f002:**
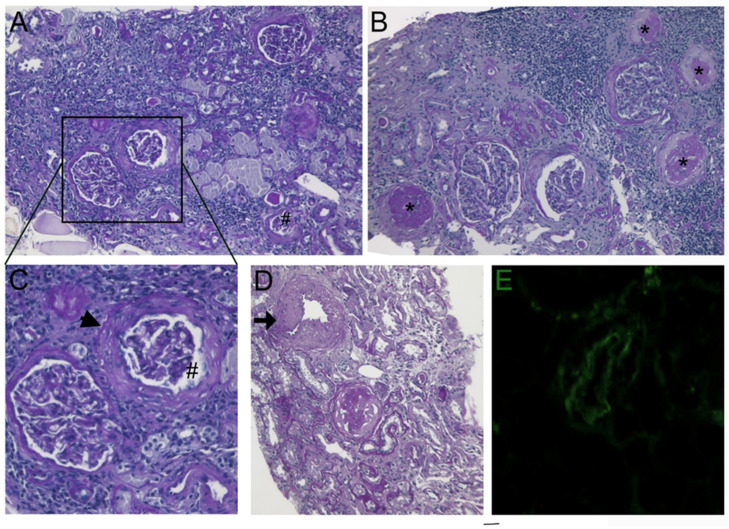
(**A**–**C**) Deflated (#) and globally sclerosed glomeruli (asterisks) associated with capsule thickening and pericapsular fibrosis (arrowhead). Atrophic tubules and interstitial fibrosis with some lymphocytic infiltrates are also present (PAS original magnification, ×100). (**D**) Medium-sized artery showed thickening of the media (arrow) (PAS original magnification, ×100). (**E**) Immunofluorescence of arteriole staining with C3 antiserum (C3 antiserum ×400).

## Data Availability

The datasets used and analyzed during the current study are available from the corresponding author upon reasonable request.

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
