# Peer review of "Novel Genetic Microvascular Dysplasia Causing Hypoperfusion of Cardiac, Renal, and Cerebral Circulation"

_jcm, 2023, doi:10.3390/jcm12227150_

Round 1

Reviewer 1 Report

Comments and Suggestions for Authors

At the begining I would like to congratulate the authors very interesting presentation of such unusual clinical case. The case report is very detailed and precisely presents the individual steps in diagnosis. The study was approved by the bioethics committee.

In my opinion, it is particulary valuable that researches managed to identify the genetic variation responsible for the inheritance of this rare defect.

The valvue of the work is increased by photo documentation and attached videos.

Author Response

We thank the reviewer for his/her comments

Reviewer 2 Report

Comments and Suggestions for Authors

The authors of the manuscript present a case of a patient who experienced diseases affecting his heart, kidneys and brain within 35 years. The authors identify some similarities in the histological picture of the heart and kidneys (the tests were performed 20 years apart). In a genetic study, they identify 3 rare heterozygous missense variants in genes that may be associated with vasculopathies and conclude that they are the cause of the observed disorders.

The proposed concept is interesting, but there is no evidence of a cause-and-effect relationship between the identified variants and the observed disorders. Therefore, it should be treated as a hypothesis.

I understand that proving the pathogenicity of variants by creating cellular models or even in silico evaluation may be beyond the scope of this work, but I believe that the authors should at least describe the clinical picture in more detail.

1. What were the results regarding common causes of TIA / stroke at the age of 65? Why was mannitol administered? What were the results of his cardiac tests at the time ? What typical cardiovascular risk factors were present? Was CMR  performed - at that time or earlier? - the authors state that the CMR was performed, but do not provide its results.

2. What was the differential diagnosis for renal involvement? How were alternative diagnoses excluded?

3. What was the family history? Have the relatives been tested and if not, why?

4. I feel not qualified to assess the quality of English, however I have some remarks. Angio-Tac is not a commonly used abbreviation. I would suggest using computed tomography angiography or CTA. What do you mean by epiaortic arteries?

5. The authors state that the variants were classified according to ACMG, but this classification is missing in the manuscript and should be included in the text. 

Author Response

The authors of the manuscript present a case of a patient who experienced diseases affecting his heart, kidneys and brain within 35 years. The authors identify some similarities in the histological picture of the heart and kidneys (the tests were performed 20 years apart). In a genetic study, they identify 3 rare heterozygous missense variants in genes that may be associated with vasculopathies and conclude that they are the cause of the observed disorders.

The proposed concept is interesting, but there is no evidence of a cause-and-effect relationship between the identified variants and the observed disorders. Therefore, it should be treated as a hypothesis.

Reply: The statement that there is no demonstration of a cause-and-effect relationship between systemic microvascular changes and identified genes mutation but just a hypothetical correlation, is now reported in the discussion section.

I understand that proving the pathogenicity of variants by creating cellular models or even in silico evaluation may be beyond the scope of this work, but I believe that the authors should at least describe the clinical picture in more detail.

  1. What were the results regarding common causes of TIA / stroke at the age of 65? Why was mannitol administered? What were the results of his cardiac tests at the time? What typical cardiovascular risk factors were present? Was CMR performed - at that time or earlier? - the authors state that the CMR was performed, but do not provide its results.

Reply: Patient clinical and hematochemical characteristics as well as cardiac magnetic resonance findings are now reported in the Case study section highlighted in bold. Mannitol or glycerol infusion was used successfully to contrast cerebral oedema and neurologic manifestations whenever other pharmacologic measures (including aspirin, nimodipine, nitrates and anticoagulants) failed. Finally, we introduced in the text a wide discussion considering the differential diagnosis with the major alternative entities.

  1. What was the differential diagnosis for renal involvement? How were alternative diagnoses excluded?

      Reply: This aspect is now analysed in the discussion.

  1. What was the family history? Have the relatives been tested and if not, why?

      Reply: We thank the reviewer for these comments. The family history was negative. Unaffected family members were not tested because they were not available for testing. This information has been added in the revised version of the manuscript.

  1. I feel not qualified to assess the quality of English, however I have some remarks. Angio-Tac is not a commonly used abbreviation. I would suggest using computed tomography angiography or CTA. What do you mean by epi-aortic arteries?

Reply: CTA replaced angio-Tac. For epi-aortic arteries it is intended carotid, vertebral and major cerebral arteries. This is now specified in the text.

  1. The authors state that the variants were classified according to ACMG, but this classification is missing in the manuscript and should be included in the text.

Reply: According with the reviewer's suggestion, we included the ACMG classification of genetic variants in the text of the revised version of the manuscript.

 6) At that time the invasive study was implemented with bi-ventricular endomyocardial biopsy

Why was a biopsy performed if the patient had normal left ventricular muscle contractility?

Reply: Decision to perform LV endomyocardial biopsy followed the angiographic evidence of  biventricular microaneurysms. In our experience one of major causes is myocardial inflammation which should be confirmed and delineated on its infectious or immune-mediated mechanism in order to provide a specific treatment. (Inflammatory Left Ventricular Microaneurysms as a Cause of Apparently Idiopathic Ventricular Tachyarrhythmias.

Cristina Chimenti Fiorella Calabrese,Gaetano Thiene,Maurizio Pieroni,Attilio Maseri and Andrea Frustaci

Originally published10 Jul 2001https://doi.org/10.1161/01.CIR.104.2.168 Circulation. 2001;104:168–173)

Reviewer 3 Report

Comments and Suggestions for Authors

1.       At that time the invasive study was implemented with bi-ventricular endomyocardial biopsy

Why was a biopsy performed if the patient had normal left ventricular muscle contractility?

2.       Microcirculatory disorders also occur in people after anthracycline chemotherapy. Please cite the article below:

Anthracycline-induced microcirculation disorders: AIM PILOT Study.

Klotzka A, Iwańczyk S, Ropacka-Lesiak M, Misan N, Lesiak M.Kardiol Pol. 2023;81(7-8):766-768. doi: 10.33963/KP.a2023.0108. Epub 2023 May 16.

3.       Very interesting article. Innovative. Possibly a new disease entity, multisystem because dependent on damage to the vascular wall.

The introduction and methodology are included but a brief summary is missing.

Author Response

  1. Microcirculatory disorders also occur in people after anthracycline chemotherapy. Please cite the article below:

Anthracycline-induced microcirculation disorders: AIM PILOT Study.

Klotzka A, Iwańczyk S, Ropacka-Lesiak M, Misan N, Lesiak M.Kardiol Pol. 2023;81(7-8):766-768. doi: 10.33963/KP.a2023.0108. Epub 2023 May 16.

 Reply: The suggested potential mechanism of microvascular damage by antracyclines administration is mentioned in the discussion and the related reference included.

  1. Very interesting article. Innovative. Possibly a new disease entity, multisystem because dependent on damage to the vascular wall.

Reply: We very much appreciate the reviewer’s consideration.

  1. The introduction and methodology are included but a brief summary is missing.

Reply: A brief summary is now provided

Round 2

Reviewer 2 Report

Comments and Suggestions for Authors

In response to the review, the authors added a lot of important information, which allows for a more accurate assessment of the presented case. However, I believe that there is still a lot of important information missing and therefore inference is difficult.

1. The authors start with description of TIA but give no details of echo, standard and Holter ECG, Doppler ultrasound at that time. Such an assessment is obligatory in search of the causes of TIA and it was probably carried out. The results should be described extensively, especially since they would also show the course of heart disease over >30 years.

Did the LV aneurysms undergo further remodeling? Maybe they were the cause of cerebral embolism? Were they causing ventricular arrhythmia? What was the left ventricular contractility at the time of TIA? Was there progression of fibrosis in LV? Were atherosclerotic plaques present in carotid arteries?

Without all this information, the hypothesis that ischemic incidents could be caused by genetic angiopathy seems unfounded.

2. The paragraph dedicated to kidney pathology is small. It seems that the microscopic image is dominated not by vascular lesions, but with glomerular sclerosis, and its causes include hyperfiltration, toxic or viral factors rather than ischemia. That's why I asked for differential diagnostics.
As a cardiologist I do not want to speak definitively, but also at this point the authors' reasoning is not fully convincing for me.

3. Although the authors admit that there is no firmly proven association between the found genetic variants and the clinical image, there are  statements about a new genetic entity in the manuscript, among others in the title, abstract and summary.
These statements, based on the description of a single patient with several  variants unknown significance, should be modified (including the title).

Author Response

  1. The authors start with description of TIA but give no details of echo, standard and Holter ECG, Doppler ultrasound at that time. Such an assessment is obligatory in search of the causes of TIA and it was probably carried out. The results should be described extensively, especially since they would also show the course of heart disease over >30 years.

Did the LV aneurysms undergo further remodeling? Maybe they were the cause of cerebral embolism? Were they causing ventricular arrhythmia? What was the left ventricular contractility at the time of TIA? Was there progression of fibrosis in LV? Were atherosclerotic plaques present in carotid arteries?

Without all this information, the hypothesis that ischemic incidents could be caused by genetic angiopathy seems unfounded.

Reply: We thank the reviewer for the ponts raised.

We specify in the text that patient admissions because of cerebral ischemia were characterized by:

  • Normal ECG and absence of cardiac arhythmias at Holter monitoring.
  • Normal cardiac dimension and biventricular function.
  • Absence of cardiac thrombi at 2D-echo and CMR.
  • Normality of carotid, vertebral and main intracerebral arteries.

  1. The paragraph dedicated to kidney pathology is small. It seems that the microscopic image is dominated not by vascular lesions, but with glomerular sclerosis, and its causes include hyperfiltration, toxic or viral factors rather than ischemia. That's why I asked for differential diagnostics.

As a cardiologist I do not want to speak definitively, but also at this point the authors' reasoning is not fully convincing for me.

Reply: The clinical and histologic renal findings strongly suggested renal microvascular osbrtuctive dysplasia wre at the base of renal pathology . These findings were similar to those observed at endomyocardial biopsy.

  1. Although the authors admit that there is no firmly proven association between the found genetic variants and the clinical image, there are statements about a new genetic entity in the manuscript, among others in the title, abstract and summary.

These statements, based on the description of a single patient with several  variants unknown significance, should be modified (including the title).

Reply: On the base of recurrent TIAs with multiple small ischemic lesions at brain magnetic resonance in the absence of cardiac thrombi and arrhythmias , preserved cardiac dimension and contractility, normality of carotid, vertebral and main cerebral arteries, we do believe that the cerebral ischemic lesions have likely the same microvascular substrate of that described at cardiac and renal biopsy.

Reviewer 3 Report

Comments and Suggestions for Authors

Thank you, all comments corrected

Author Response

Thank you for your consideration